# HEAR: Hearing Enhanced Audio Response for Video-grounded Dialogue

**Sunjae Yoon**[†]    **Dahyun Kim**[†]    **Eunseop Yoon**[†]    **Hee Suk Yoon**[†]
**Junyeong Kim**[‡]    **Chang D. Yoo**[†*]

[†]Korea Advanced Institute of Science and Technology (KAIST)
[‡]Chung-Ang University
{sunjae.yoon,dahyun.kim,esyoon97,hskyoon,cd_yoo}@kaist.ac.kr
junyeongkim@cau.ac.kr

## Abstract

Video-grounded Dialogue (VGD) aims to answer questions regarding a given multi-modal input comprising video, audio, and dialogue history. Although there have been numerous efforts in developing VGD systems to improve the quality of their responses, existing systems are competent only to incorporate the information in the video and text and tend to struggle in extracting the necessary information from the audio when generating appropriate responses to the question. The VGD system seems to be deaf, and thus, we coin this symptom of current systems' ignoring audio data as a deaf response. To overcome the deaf response problem, Hearing Enhanced Audio Response (HEAR) framework is proposed to perform sensible listening by selectively attending to audio whenever the question requires it. The HEAR framework enhances the accuracy and audibility of VGD systems in a model-agnostic manner. HEAR is validated on VGD datasets (i.e., AVSD@DSTC7 and AVSD@DSTC8) and shows effectiveness with various VGD systems.

## 1 Introduction

One of the desiderata in our vision-language community is to build conversational agents that can look, listen, think and speak as humans. These agents can potentially be deployed in various subsections of society, including education, security, entertainment, and visual or other impairments. To promote natural conversation between humans and the conversational agents, the Video-grounded Dialogue (VGD) task (Alamri et al., 2019) has been designed, aiming to respond to the questions regarding a given multimodal input comprising video, audio, and dialogue history. To be specific, as shown in Figure 1, given video $V$, audio $U$, dialogue history composed of caption $C$ and past rounds of Q&A pairs $H = \{C, (Q^1, A^1), ..., (Q^{r-1}, A^{r-1})\}$, and current $r$-th

_____________
*Corresponding author

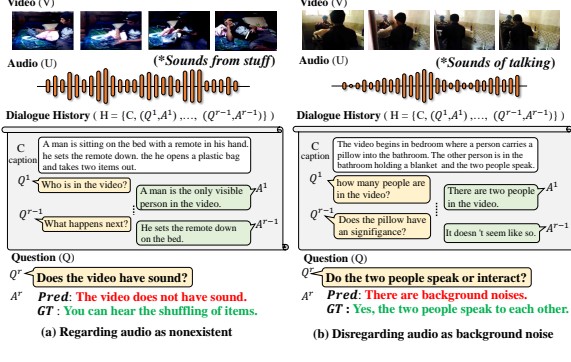

Figure 1: Current VGD system's deaf responses on questions about audio: (a) Audio is considered not present and (b) Audio is disregarded as background noise.

round question $Q^r$, VGD system is expected to answer in free-form $A^r$ to the question $Q^r$. For this VGD task, multi-modal interaction has been a popular solution, including transformer (Vaswani et al., 2017), where many studies concerning modality interactions are performed to boost performance and improve the efficiency of VGD systems.

Unfortunately, these multi-modal interactions focus only on finding the joint representations between video and language. As a consequence, current agents tend to ignore audio in generating the response. This symptom of ignoring input audio in responding to the question will be referred to as the 'deaf response'. Figure 1 represents examples of deaf responses of existing VGD systems (Yoon et al., 2022c; Li et al., 2021a). To the question "Does the video have sound?" in Figure 1 (a), the system is not able to recognize the input audio: It responds as though the audio is not present. Furthermore, even when the system does recognize the existence of audio, it lacks the capability to decipher the information within the audio accurately. This is evident in Figure 1 (b) where all the sounds of people talking are disregarded as background noise, resulting in incorrect responses.

Our experimental evidence in Figure 2 also

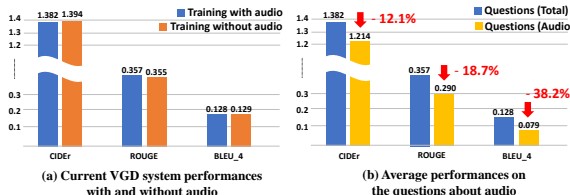

Figure 2: Current VGD systems' performances on AVSD dataset (validation): (a) Response performances according to training with and without audio, (b) Average performance drops on the questions about audio.

shows that the current VGD systems cannot incorporate audio information when responding. Figure 2 (a) shows the performance of response (*i.e.*, CIDEr (Vedantam et al., 2015), ROUGE-L (Lin, 2004), BLEU (Papineni et al., 2002)) of current VGD systems according to training with and without the audio: There are very little differences in performances between the two cases. Some metrics even show a slightly higher performance when trained without audio. Furthermore, as shown in Figure 2 (b), when investigating the responses to audio-related questions[1], their performances are noticeably lower compared to the overall response performances. Therefore, existing VGD systems tend to ignore the audio and suffer from the deaf response, which leads to incorrect answers, especially to questions pertinent to audio.

To overcome the deaf response problem, we propose Hearing Enhanced Audio Response (HEAR) framework, which allows the VGD systems to sensibly listen to audio according to the meaning of the question and perform enhanced listening to the audio. Thus, HEAR incorporates (1) Sensible Audio Listening (SAL) that selectively attends to audio according to a sensible decision of whether to focus on audio or not and (2) Reconstructive Listening Enhancement (RLE) that improves the audibility via establishing a reconstruction upper bound to connect audio with its surrounding information. For the sensible decision in SAL, we introduce two technical contributions: (1) Keyword-based Audio Sensing and (2) Semantic Neural Estimator. HEAR is applied on current runner models (Hori et al., 2019a; Yoon et al., 2022c; Li et al., 2021b) in a model-agnostic manner, where the effectiveness is validated on VGD dataset (*i.e.*, AVSD@DSTC7, AVSD@DSTC8) with steady performance gains on natural language generation metrics.

---

[1] We select the questions that contain words related to audio such as 'sound', 'speech', and 'noise'.

## 2 Related works

### 2.1 Video-grounded Dialogues

Visual Question Answering (VQA) (Antol et al., 2015; Wang et al., 2022) has been one of the proxy tasks to evaluate the multi-modal understanding of vision-language systems. In light of recent advancements in generative models in natural language processing (Devlin et al., 2018; Radford et al., 2018), VQA has evolved into a more general format of answering as video-grounded dialogue (VGD) (Alamri et al., 2019), where VGD aims to generate open-ended answer sentences from a question by referring to several input modalities (*i.e.*, video, audio, and dialogue history). Many multi-modal interactions have been proposed, where various attention mechanisms (Sanabria et al., 2019; Le et al., 2019) have been devised to perform cross-modal interactions. To boost performances, transformer-based VGD systems (Li et al., 2021b) are utilized on top of large-scale pre-trained language models (Radford et al., 2019; Raffel et al., 2020). Another immense challenge is keeping track of extended dialogue context, and video, where memory networks (Lin et al., 2019; Xie and Iacobacci, 2020) and multi-step attention (Chu et al., 2020) were introduced to efficiently store the video and long episodic dialogue. Graph representations (Kim et al., 2021; Pham et al., 2022; Le et al., 2021) were also popular solutions for holding semantic commonalities between the dialogue and video. There have been novel structures (Hori et al., 2019b; Le et al., 2022) to enhance the multi-modal representation and frameworks (Le and Chen, 2020; Lee et al., 2020; Yoon et al., 2022c) to improve the quality of responses in terms of bias or word selection. As such, many advances have been made in the multi-modal understanding of VGD systems, but mainly between video and language. Thus, the VGD system's understanding of audio is still far from satisfactory. To this end, we first contribute to improving the 'listening' ability of VGD system.

## 3 Task Definition

Video-grounded Dialogue (Alamri et al., 2019) (VGD) task aims to generate open-ended answer sentences to a question regarding multimodal inputs composed of video, audio, and dialogue history. To build a formal task definition of VGD, a system takes tuples $(v, u, h, q^r)$ as input and decodes answer sentence $a^r$, where $v$ is video, $u$ is au-

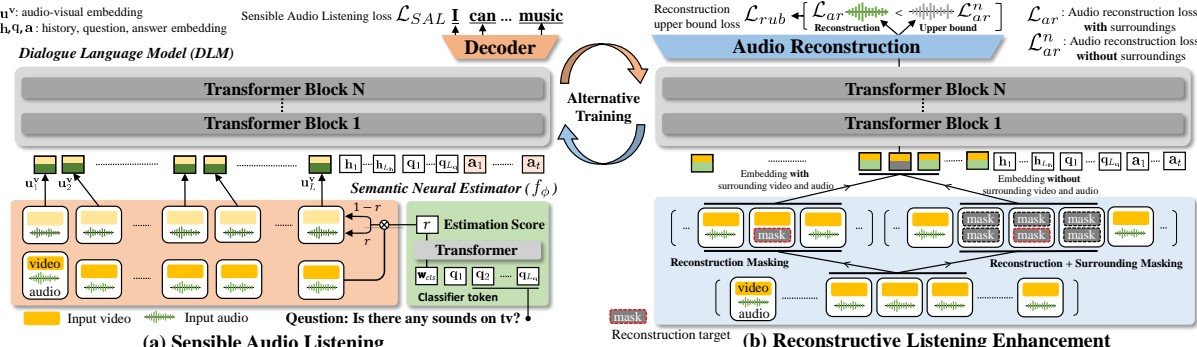

Figure 3: Illustration of Hearing Enhanced Audio Response Framework (HEAR) for video-grounded dialogue. HEAR performs sensible listening via (a) Sensible Audio Listening that selectively attends to audio corresponding to a given question and improves audibility via (b) Reconstructive Listening Enhancement that enhances audio representations by establishing a reconstruction upper bound to connect audio with its surrounding information.

dio, $h$ is dialogue history, and $q^r$ is a question asked at current round $r \in \{1, \cdots, R\}$. The dialogue history $h = \{c, (q^1, a^1), \cdots, (q^{r-1}, a^{r-1})\}$ is composed of caption $c$ to summarize the video and a set of question-answer pairs in previous rounds. For training VGD system, next-word prediction is performed, where the system predicts $t$-th answer word token $a_t^r$ from the inputs of tuples $(v, u, h, q^r)$ and partial answer word tokens $a_{<t}^r$ before $t$-th.

## 4  Hearing Enhanced Audio Response

Figure 3 illustrates Hearing Enhanced Audio Response (HEAR) framework designed to enhance Dialogue Language Model (DLM)[2] in terms of two functionalities on audio: (1) Sensibility that selectively attends on audio according to the meaning of the question and (2) Audibility that performs enhanced listening to input audio. For sensibility, we propose Sensible Audio Listening (SAL) in Figure 3 (a) which trains the DLM to respond to a question by selectively weighting to audio corresponding to the audio-relatedness of the question. For audibility, we devise Reconstructive Listening Enhancement (RLE) in Figure 3 (b) which enhances audio representations by establishing a reconstruction upper bound to connect audio with its surrounding information. We alternately train DLM with SAL and RLE to fully utilize video and audio modalities.

### 4.1  Input representations

We formally define input feature representations of $v$, $h$, $q^r$, and $a^r$ by embedding them into $d$-dimensional space. For the video embedding, we use I3D model (Carreira and Zisserman, 2017) pre-

trained on the Kinetics dataset (Kay et al., 2017) to get 4096-dimensional video features $\mathbf{v} \in \mathbb{R}^{L \times 4096}$ composed of rgb and optical-flow features, where $L$ is the number of video frames. For the audio embedding, we use VGGish model (Hershey et al., 2017) pre-trained on the AudioSet dataset (Gemmeke et al., 2017) to get 128-dimensional audio features $\mathbf{u} \in \mathbb{R}^{L \times 128}$, where the $L$ is the number of audio features[3]. The aforementioned video and audio features are concatenated along the feature dimension axis and embedded into $d$-dimensional space as audio-visual features $\mathbf{u^v}$ as given below:

$$\mathbf{u^v} = [\mathbf{u}||\mathbf{v}]W \in \mathbb{R}^{L \times d}, \tag{1}$$

where $W \in \mathbb{R}^{(128+4096) \times d}$ is $d$-dimensional embbeder and $[\cdot||\cdot]$ denotes concatenation.

For the text features, we tokenize all the text inputs (*i.e.*, $h, q^r, a^r$) into a series of WordPieces (Wu et al., 2016) using the T5 Transformer (Raffel et al., 2020), such that word token representations are obtained on top of relative positional embeddings and a layer normalization (Ba et al., 2016). Thus, the formal definitions of text features are as follows: history $\mathbf{h} \in \mathbb{R}^{L_\mathbf{h} \times d}$, question $\mathbf{q} \in \mathbb{R}^{L_\mathbf{q} \times d}$, and answer $\mathbf{a} \in \mathbb{R}^{L_\mathbf{a} \times d}$, where $L_\mathbf{h}, L_\mathbf{q}$, and $L_\mathbf{a}$ are the numbers of tokens of each text[4].

### 4.2  Dialogue Language Model

Our proposed HEAR framework is performed in a model-agnostic manner, such that we first define Dialogue Language Model (DLM) as a general VGD system. For input audio, video, and texts (*i.e.*, history and question), DLM is trained

---

[2]Here, we refer to the general 'VGD system' as DLM.

[3]Sample rate is modulated to be aligned with video frames.
[4]We delete superscripts $r$ in the notations for the simplicity.

to generate next-word tokens for answer sentences $a^r = \{a_1^r, \cdots, a_m^r\}$ under cross-entropy loss as:

$$\mathcal{L}_{DLM}(\theta) = -\log \prod_{t=1}^{m} P_\theta(a_t^r | \mathbf{u^v}, \mathbf{h}, \mathbf{q}, \mathbf{a}_{<t}), \quad (2)$$

where $\theta$ is learnable weights and $m$ is the number of word tokens in the answer sentence. In the following, our proposed SAL and RLE improve the DLM's sensibility and audibility for correctly responding to questions about audio.

### 4.3 Sensible Audio Listening

Sensible Audio Listening (SAL) is devised to determine whether the VGD system should listen to the audio to answer a given question. If the SAL determines that listening is required, audio is processed to be more weighted in the response. To ensure sensible decision-making within SAL, we introduce two technical decision rules: (1) Keyword-based Audio Sensing and (2) Semantic Neural Estimator.

**Keyword-based Audio Sensing.** In many cases, unlike general questions, audio-related questions contain specific keywords (*e.g.*, 'sound', 'speech', 'listen') that indicate that the question concerns audio. Therefore, we conduct an empirical investigation of these keywords. If any of these keywords are present in a question, SAL identifies it as an audio-related question. In such cases, we mask the video features in the inputs, directing more attention toward the audio component as given below:

$$\mathbf{u^v}(q) = \begin{cases} [\mathbf{u}||\mathbf{v}]W & \forall w_q \notin \mathbf{W}_{key} \\ [\mathbf{u}||\mathbf{m_v}]W & \text{otherwise,} \end{cases} \quad (3)$$

where $w_q$ is all word tokens in the question $q$ and $\mathbf{W}_{key} = \{\text{'sound', 'speech', 'listen', \dots}\}$ is keyword set[5] to investigate audio-related questions. $\mathbf{m_v} \in \mathbb{R}^{L \times 4096}$ is zero padding on video. Here, we do not perform any masking when the question is not an audio-related question (*i.e.*, $\forall w_q \notin \mathbf{W}_{key}$), because other questions excluding audio-related questions do not show high reliance on a specific modality. Thus, $\mathbf{u^v}(q) \in \mathbb{R}^{L \times d}$ is sensible audio-visual features that make DLM selectively focus on audio for responding to the audio-related questions.

**Semantic Neural Estimator.** Figure 4 (a) illustrates audio-related questions identified by

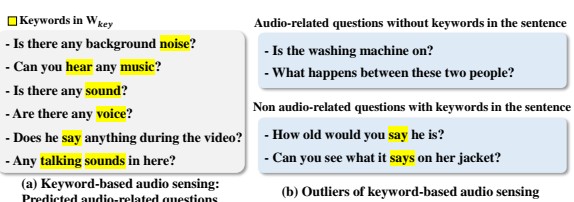

Figure 4: Examples of questions: (a) Predicted audio-related questions by keyword-based audio sensing and (b) Outliers of keyword-based audio sensing.

keyword-based audio sensing. While the keyword-based approach effectively identifies the questions that directly require information about audio, we were still able to identify outliers that were not accurately identified in Figure 4 (b). Although they do not contain keywords, the meanings of the questions are related to audio. Therefore, we further devise a semantic neural estimator $f_\phi$ that identifies the audio-related question based on the meaning of sentences. The semantic neural estimator as a BERT-based classifier takes $\{w_{\text{cls}}, w_q\}$ to form an input instance, where $w_{\text{cls}}$ is a token for classification and the prediction target of $w_{\text{cls}}$ is $y \in \{y_0, y_1\}$ denoting that $y_1 = 1$ is audio-related question and $y_0 = 0$ is the other question. We first train the $f_\phi$ with training weights $\phi$ under L2 loss[6] given as $\mathbb{E}_D(y - \hat{y})^2$, where $D$ is the dataset, $\hat{y} = f_\phi(w_{\text{cls}}, w_q)$ is prediction score of the sigmoid (*i.e.*, $0 < \hat{y} < 1$). For labeling the $y$, we first include the predictions by keyword-based audio sensing as a noisy label. We further include $y_0$ with (1) word-wise random shuffled versions of audio-related questions and (2) random swapping versions of non-audio questions using keyword $\mathbf{W}_{key}$. This prevents $f_\phi$ from simply predicting based on the keywords and makes more focus on the meaning of the audio-related questions in $y_1$. After training $f_\phi$, we calibrate the audio and video features according to the estimation of $f_\phi$ as:

$$\mathbf{u^v}(q) = [r \times \mathbf{u}||(1 - r) \times \mathbf{v}]W, \quad (4)$$

where $r = f_\phi(w_{\text{cls}}, w_q) \in \mathbb{R}$ is estimation score about audio-related question and $\mathbf{u^v}(q) \in \mathbb{R}^{L \times d}$ is our final sensible audio-visual features for SAL. Based on the sensible audio-visual feature $\mathbf{u^v}(q)$ from SAL, we train Dialogue Language Model (DLM) with cross-entropy loss as given below:

$$\mathcal{L}_{SAL}(\theta) = -\log \prod_{t=1}^{m} P_\theta(a_t^r | \mathbf{u^v}(q), \mathbf{h}, \mathbf{q}, \mathbf{a}_{<t}). \quad (5)$$

---

[5]Appendix provides all the keywords that we used.

[6]Regularization is also used to mitigate training imbalance.

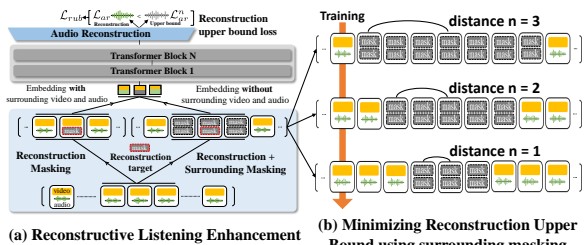

(a) Reconstructive Listening Enhancement

(b) Minimizing Reconstruction Upper Bound using surrounding masking

Figure 5: Illustrations of surrounding masking. The distance $n$ decides the extent of surrounding masking.

### 4.4 Reconstructive Listening Enhancement

For a better understanding of audio, it is crucial to improve audio representations based on a common understanding of a scene in a video. Therefore, our proposed Reconstructive Listening Enhancement (RLE) performs masked audio reconstruction by referring to surrounding information (*i.e.*, video and audio adjacent to masked audio), which enhances the common embedding with its surroundings. Especially to perform effective enhancement in regions closer to the masked target, we propose Reconstruction Upper Bound in the following.

**Audio Reconstruction.** Audio reconstruction aims to predict masked audio based on observations of their surrounding audio and other modalities (*i.e.*, video and text). We randomly select input audio with a probability of $p$ (*e.g.*, $p$=10%) as $\mathbf{u}_m$, where $\mathbf{u}_m \in \mathbb{R}^{M \times 128}$ is the target audio features to be masked, $m$ is the set of indices for masking, and $M$ is the number of indices. We also define $\mathbf{u}_{\backslash m} \in \mathbb{R}^{L \times 128}$ as surrounding audio features including masked features of zero vectors in the indices $m$. We introduce audio reconstruction loss $\mathcal{L}_{ar}$ for DLM to reconstruct target audio features $\mathbf{u}_m$ from the inputs of $\{\mathbf{u}_{\backslash m}, \mathbf{v}, \mathbf{h}, \mathbf{q}\}$ as:

$$\mathcal{L}_{ar}(\theta) = h_\theta(\mathbf{u}_m | \mathbf{u}_{\backslash m}, \mathbf{v}, \mathbf{h}, \mathbf{q}). \quad (6)$$

$h_\theta(\mathbf{u}_m | \mathbf{u}_{\backslash m}, \mathbf{v}, \mathbf{h}, \mathbf{q}) = \sum_{i=1}^{M} ||\mathbf{u}_m^{(i)} - \tilde{\mathbf{u}}_m^{(i)}||_2^2$ is l2 regression, where $\tilde{\mathbf{u}}_m \in \mathbb{R}^{M \times 128}$ is reconstructed audio on the masked indices $m$ from the output of multi-layer perceptron on top of DLM encoder.

**Reconstruction Upper Bound.** The audio reconstruction should be based on an understanding of surrounding semantics rather than simply memorizing audio. To facilitate this, we propose Reconstruction Upper Bound (RUB) which establishes an inequality condition to ensure enhanced reconstruction when the surrounding information is provided

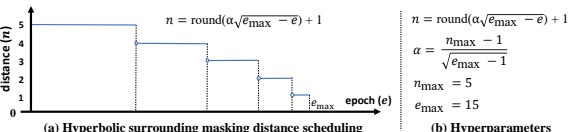

(a) Hyperbolic surrounding masking distance scheduling

(b) Hyperparameters

Figure 6: Hyperbolic function for distance scheduling.

compared to when it is not given as given below:

$$\mathcal{L}_{ar}(\theta) < h_\theta(\mathbf{u}_m | \mathbf{u}_{\backslash m}^n, \mathbf{v}^n, \mathbf{h}, \mathbf{q}), \quad (7)$$

where the audio reconstruction loss based on surroundings $\mathcal{L}_{ar}(\theta) = h_\theta(\mathbf{u}_m | \mathbf{u}_{\backslash m}, \mathbf{v}, \mathbf{h}, \mathbf{q})$ should be lower than the reconstruction loss without surroundings $\mathcal{L}_{ar}^n(\theta) = h_\theta(\mathbf{u}_m | \mathbf{u}_{\backslash m}^n, \mathbf{v}^n, \mathbf{h}, \mathbf{q})$ (*i.e.*, upper bound). As shown in Figure 5 (a), $\mathbf{u}_{\backslash m}^n, \mathbf{v}^n$ are audio and video features that further mask out surroundings up to feature distance $n$ (*e.g.*, $n = 3$) from the target audio features. To train DLM to conform this inequality condition, we calculate a reconstruction upper bound loss $\mathcal{L}_{rub}$ in a format of ranking loss with margin $\delta$ as given below:

$$\mathcal{L}_{rub}(\theta, n) = \max(\mathcal{L}_{ar}(\theta) - \mathcal{L}_{ar}^n(\theta) + \delta, 0). \quad (8)$$

After that, we iteratively minimize the upper bound $\mathcal{L}_{ar}^n$ in a way of narrowing the surrounding masking with distance $n$[7] as shown in Figure 5 (b). Here, we construct masking distance scheduling $n = g(e)$ according to taring epoch $e$ to select progressively lower $n$ from higher $n$[8]. Therefore, the final objective of RLE is formulated as below:

$$\min_{\theta, g \sim G} \mathcal{L}_{ar}(\theta) + \mathcal{L}_{rub}(\theta, g(e)), \quad (9)$$

where $G : \mathbb{R}^+ \mapsto \mathbb{R}^+$ is a set of scheduling functions and we select hyperbolic function[9] as $g(e) = \text{round}(\alpha\sqrt{e_{\max} - e}) + 1$ as depicted in Figure 6 to give stable decreasing of $n$ with $\alpha = \frac{n_{\max} - 1}{\sqrt{e_{\max} - 1}}$ and $n_{\max}$=5, $e_{\max}$=15 are maximum distance and epoch. Therefore, the final RLE loss is the summation of two losses: $\mathcal{L}_{RLE} = \mathcal{L}_{ar} + \mathcal{L}_{rub}$.

### 4.5 Optimization and Inference

As shown in Figure 3, we train DLM by alternately optimizing $\mathcal{L}_{SAL}$ and $\mathcal{L}_{RLE}$ to allow audio and

---

[7]As the surrounding masking is reduced, the DLM can perform the reconstruction better, thus $\mathcal{L}_{ar}^n$ decreases.

[8]Empirically found that low $n$ in early training (*e.g.*, S(e) = 1) ruins training stability, as $\mathcal{L}_{ar}, \mathcal{L}_{ar}^n$ were almost similar.

[9]See more scheduling functions in Appendix B.2.

| AVSD@DSTC7 | | | | | | | |
|---|---|---|---|---|---|---|---|
| Methods | B1 | B2 | B3 | B4 | M | R | C |
| EE-DMN (Lin et al., 2019) | 0.641 | 0.493 | 0.388 | 0.310 | 0.241 | 0.527 | 0.912 |
| JMAN (Chu et al., 2020) | 0.667 | 0.521 | 0.413 | 0.334 | 0.239 | 0.533 | 0.941 |
| CMU (Sanabria et al., 2019) | 0.718 | 0.584 | 0.478 | 0.394 | 0.267 | 0.563 | 1.094 |
| COST (Pham et al., 2022) | 0.723 | 0.589 | 0.483 | 0.400 | 0.266 | 0.561 | 1.085 |
| MSTN (Lee et al., 2020) | - | - | - | 0.377 | 0.275 | 0.566 | 1.115 |
| JSTL (Hori et al., 2019b) | 0.727 | 0.593 | 0.488 | 0.405 | 0.273 | 0.566 | 1.118 |
| MTN* (Le et al., 2019) | 0.731 | 0.597 | 0.490 | 0.406 | 0.271 | 0.564 | 1.127 |
| MTN-P (Le and Chen, 2020) | 0.750 | 0.619 | 0.514 | 0.427 | 0.280 | 0.580 | 1.189 |
| VGNMN (Le et al., 2022) | - | - | - | 0.429 | 0.278 | 0.578 | 1.188 |
| SCGA (Kim et al., 2021) | 0.745 | 0.622 | 0.517 | 0.430 | 0.285 | 0.578 | 1.201 |
| RLM (Li et al., 2021b) | 0.765 | 0.643 | 0.543 | 0.459 | 0.294 | 0.606 | 1.308 |
| PDC (Le et al., 2021) | 0.770 | 0.653 | 0.539 | 0.449 | 0.292 | 0.606 | 1.295 |
| THAM (Yoon et al., 2022c) | 0.778 | 0.654 | 0.549 | 0.468 | 0.308 | 0.619 | 1.335 |
| **HEAR (SAL)** | 0.784 | 0.658 | 0.551 | 0.471 | 0.309 | 0.619 | 1.347 |
| **HEAR (SAL + RLE)** | **0.791** | **0.662** | **0.558** | **0.472** | **0.312** | **0.622** | **1.376** |
| AVSD@DSTC8 | | | | | | | |
| MDMN (Xie and Iacobacci, 2020) | - | - | - | 0.296 | 0.214 | 0.496 | 0.761 |
| JMAN (Chu et al., 2020) | 0.645 | 0.504 | 0.402 | 0.324 | 0.232 | 0.521 | 0.875 |
| STSGR (Geng et al., 2021) | - | - | - | 0.357 | 0.267 | 0.553 | 1.004 |
| MSTN (Lee et al., 2020) | - | - | - | 0.385 | 0.270 | 0.564 | 1.073 |
| COST (Pham et al., 2022) | 0.695 | 0.559 | 0.465 | 0.382 | 0.278 | 0.574 | 1.051 |
| MTN-P (Le and Chen, 2020) | 0.701 | 0.587 | 0.494 | 0.419 | 0.263 | 0.564 | 1.097 |
| SCGA (Kim et al., 2021) | 0.711 | 0.593 | 0.497 | 0.416 | 0.276 | 0.566 | 1.123 |
| RLM (Li et al., 2021b) | 0.746 | 0.626 | 0.528 | 0.445 | 0.286 | 0.598 | 1.240 |
| PDC (Le et al., 2021) | 0.749 | 0.629 | 0.528 | 0.439 | 0.285 | 0.592 | 1.201 |
| THAM (Yoon et al., 2022c) | 0.764 | 0.641 | 0.538 | 0.455 | 0.301 | 0.610 | 1.304 |
| **HEAR (SAL)** | 0.767 | 0.646 | 0.541 | 0.459 | 0.302 | 0.612 | 1.324 |
| **HEAR (SAL + RLE)** | **0.777** | **0.656** | **0.553** | **0.465** | **0.307** | **0.618** | **1.359** |

Table 1: Experimental results on AVSD@DSTC7 (test) and AVSD@DSTC8 (test) dataset. (B: BLEU, M: METEOR, R: ROUGE-L, C: CIDEr, *: reported in (Kim et al., 2021)). Human Evaluation is also presented in Appendix C.

video to be freely utilized for their purposes. Therefore, $\mathcal{L}_{SAL}$ is optimized when the training iteration number is odd, and $\mathcal{L}_{RLE}$ does in the even number:

$$\mathcal{L}_{HEAR} = \begin{cases} \mathcal{L}_{SAL} & \text{iteration} = odd \\ \mathcal{L}_{RLE} & \text{iteration} = even, \end{cases} \quad (10)$$

Here, RLE is only used for training. In an inference, The DLM generates an answer with SAL.

## 5 Experiments

### 5.1 Datasets

**AVSD@DSTC7 and AVSD@DSTC8.** (Audio Visual Scene-Aware Dialog) (Alamri et al., 2019; Hori et al., 2020) is a popular benchmark for VGD task, where all videos include their summary captions and dialogue composed of 10 pairs of ques-

tion and answer. The videos are collected from Charades (Sigurdsson et al., 2016) dataset, which contains natural human activities. AVSD is released at Dialogue System Technology Challenge (DSTC) 7 and 8, where AVSD@DSTC7 contains $7,659$, $1,787$, and $1,710$ dialogues for training, validation and test, but AVSD@DSTC8 only provides $1,710$ dialogues for the test. For the test set, 6 reference answers are available for accurate validation.

### 5.2 Metrics

We follow official metrics for AVSD benchmark (*i.e.*, BLEU, METEOR (Banerjee and Lavie, 2005), ROUGE-L, CIDEr) provided by AVSD challenge organizers[10], where they compute words overlaps between predicted sentence and reference answer.

---

[10]github.com/dialogtekgeek/DSTC8-AVSD_official

| AVSD@DSTC7 | | | | | | | |
|---|---|---|---|---|---|---|---|
| Methods | B1 | B2 | B3 | B4 | METEOR | ROUGE-L | CIDEr |
| MTN† (Le et al., 2019) | 0.357 | 0.241 | 0.173 | 0.128 | 0.162 | 0.355 | 1.249 |
| **MTN† + HEAR** | 0.382 | 0.264 | 0.193 | 0.144 | 0.179 | 0.374 | 1.314 |
| RLM (Li et al., 2021b) | 0.765 | 0.643 | 0.543 | 0.459 | 0.294 | 0.606 | 1.308 |
| **RLM + HEAR** | 0.787 | 0.660 | 0.556 | 0.467 | 0.308 | 0.619 | 1.358 |
| T5RLM (Yoon et al., 2022c) | 0.767 | 0.644 | 0.542 | 0.461 | 0.296 | 0.608 | 1.311 |
| **T5RLM + HEAR** | **0.791** | **0.662** | **0.558** | **0.472** | **0.312** | **0.622** | **1.376** |
| AVSD@DSTC8 | | | | | | | |
| MTN⋆ (Le et al., 2019) | 0.689 | 0.571 | 0.470 | 0.404 | 0.251 | 0.551 | 1.049 |
| **MTN⋆ + HEAR** | 0.714 | 0.596 | 0.496 | 0.421 | 0.271 | 0.569 | 1.121 |
| RLM (Li et al., 2021b) | 0.746 | 0.626 | 0.528 | 0.445 | 0.286 | 0.598 | 1.240 |
| **RLM + HEAR** | 0.772 | 0.651 | **0.554** | 0.462 | 0.303 | 0.617 | 1.323 |
| T5RLM (Yoon et al., 2022c) | 0.749 | 0.631 | 0.529 | 0.445 | 0.290 | 0.600 | 1.263 |
| **T5RLM+ HEAR** | **0.777** | **0.656** | 0.553 | **0.465** | **0.307** | **0.618** | **1.359** |

Table 2: Experimental results on AVSD@DSTC7 (test) and AVSD@DSTC8 (test) for applying HEAR on VGD runner models (B1: BLEU1, ⋆: reconstruction-based results, †: single reference results).

| SAL | | RLE | | BLEU1 | ROUGE-L | CIDEr |
|---|---|---|---|---|---|---|
| k | s | $\mathcal{L}_{ar}$ | $\mathcal{L}_{rub}$ | | | |
| | | | | 0.302 | 0.348 | 1.367 |
| ✓ | | | | 0.310 | 0.355 | 1.427 |
| | ✓ | | | 0.316 | 0.366 | 1.434 |
| | ✓ | ✓ | | 0.329 | 0.375 | 1.467 |
| | ✓ | | ✓ | 0.311 | 0.357 | 1.415 |
| ✓ | ✓ | | ✓ | **0.341** | **0.384** | **1.492** |

Table 3: Ablation studies of model variants on AVSD@DSTC7 (validation, single reference). k: Keyword-based Audio Sensing, s: Semantic Neural Estimator, $\mathcal{L}_{ar}$: Audio Reconstruction, $\mathcal{L}_{rub}$: Reconstruction Upper Bound

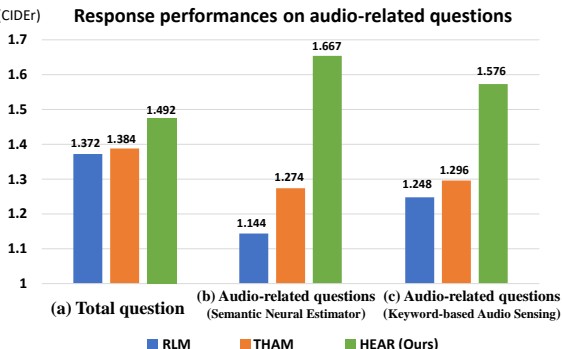

Figure 7: VGD systems' response performances on audio-related questions of AVSD@DSTC7 (validation): (a) Total questions, (b) Audio-related questions predicted by Semantic Neural Estimator (questions with estimation score $r > 0.7$), (c) Audio-related questions predicted by Keyword-based Audio Sensing.

### 5.3 Results on AVSD benchmark

Table 1 summarizes the experimental results of HEAR on the AVSD@DSTC7 and AVSD@DSTC8. HEAR shows state-of-the-art performances on all the metrics compared to previous works (*i.e.*, please refer to Related Work for their detailed captions.). Our baseline DLM is T5 Transformer (Raffel et al., 2020), which is the same baseline (*i.e.*, T5RLM) of THAM (Yoon et al., 2022c), but here, our proposed SAL shows more gains, and further improvements are also shown by applying RLE. As our proposed HEAR is performed in a model-agnostic manner, we also validate other VGD models with HEAR in Table 2. We utilize the public codes and papers for MTN, RLM, and T5RLM, where steady gains in all metrics are shown for all the models.

### 5.4 Ablation Study

Table 3 summarizes the ablative results of the proposed modules in HEAR framework. The first section of Table 3 is about our base DLM performances. In the second section, for the variants of SAL, smaller gains are obtained when only using keyword-based audio sensing. We think that using it alone was not beneficial in answering some audio-related questions that require referencing the videos, as this method unconditionally screens out video features as long as the given questions are related to audio. In the case of RLE, the audio reconstruction loss $\mathcal{L}_{ar}$ generally gives a positive effect on the system performances. However, the recon-

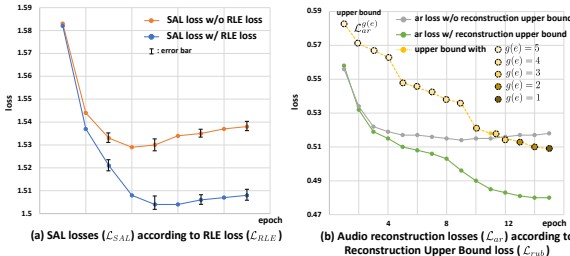

(a) SAL losses ($\mathcal{L}_{SAL}$) according to RLE loss ($\mathcal{L}_{RLE}$)

(b) Audio reconstruction losses ($\mathcal{L}_{ar}$) according to Reconstruction Upper Bound loss ($\mathcal{L}_{rub}$)

Figure 8: Ablative results on validation losses according to epochs: (a) SAL loss $\mathcal{L}_{SAL}$ with and without RLE loss $\mathcal{L}_{RLE}$, (b) Audio reconstruction loss $\mathcal{L}_{ar}$ with and without reconstruction upper bound loss $\mathcal{L}_{rub}$.

struction upper bound loss $\mathcal{L}_{rub}$ becomes effective only when the $\mathcal{L}_{ar}$ and $\mathcal{L}_{rub}$ are used together. This denotes that the ranking loss in the $\mathcal{L}_{rub}$ is founded on well-reconstructed audio features.

One may wonder if HEAR really helps answer audio-related questions. Figure 7 shows the response performances in terms of VGD systems including HEAR according to questions. HEAR performs new state-of-the-art performances on top of previous runner models, but it is also meaningful that the gains are mainly obtained from improving responses to audio-related questions. From the results in Figure 7 (b), (c) and Table 2, it is concluded that HEAR framework can be applied to any VGD system, where it exclusively contributes to improving systems' audibility and generating correct responses to given audio-related questions.

Furthermore, we found that our proposed RLE also contributes to the efficient learning of sensible audio listening (*i.e.*, optimizing $\mathcal{L}_{SAL}$). Figure 8 (a) shows validation losses ($\mathcal{L}_{SAL}$) according to training epochs, where it can be seen that the $\mathcal{L}_{SAL}$ is further reduced when RLE loss ($\mathcal{L}_{RLE}$) is applied. Figure 8 (b) shows the ablation studies of audio reconstruction loss $\mathcal{L}_{ar}$ with and without reconstruction upper bound $\mathcal{L}_{rub}$. The yellow curve shows the upper bound loss $\mathcal{L}_{ar}^{g(e)}$ for $\mathcal{L}_{rub}$, where the it decreases according to our masking distance scheduling $g(e)$[11]. The green curve denotes the $\mathcal{L}_{ar}$ with $\mathcal{L}_{rub}$ and it shows further optimizing compared to without $\mathcal{L}_{rub}$ as the $\mathcal{L}_{ar}^{g(e)}$ decreases. This denotes that neural networks can be further optimized according to the training epochs by calibrating their training objectives, which is also validated in other multi-modal systems (Yoon et al., 2023; Zheng et al., 2022) in other ways.

---

[11]We use a hyperbolic curve for $g(e)$. Appendix also contains more diverse surrounding masking scheduling.

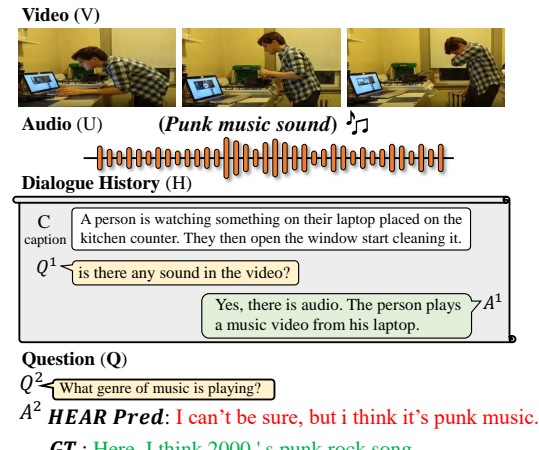

Video (V)

Audio (U)    (*Punk music sound*) 🎵

Dialogue History (H)

C caption: A person is watching something on their laptop placed on the kitchen counter. They then open the window start cleaning it.

$Q^1$: is there any sound in the video?

$A^1$: Yes, there is audio. The person plays a music video from his laptop.

Question (Q)

$Q^2$: What genre of music is playing?

$A^2$ **HEAR Pred**: I can't be sure, but i think it's punk music.

**GT** : Here, I think 2000 ' s punk rock song.

Figure 9: HEAR responses to an audio-related question.

| Questions | K | S | H |
|---|---|---|---|
| (a) *Can you hear any sounds?* | T | 0.99 | T |
| (b) *Do they speak to each other?* | T | 0.99 | T |
| (c) *Who is outside the door?* | F | 0.77 | T |
| (d) *Is the vacuum cleaner working?* | F | 0.74 | T |
| (e) *Can you tell where he goes?* | T | 0.32 | F |
| (f) *What color is his hair?* | F | 0.01 | F |

Table 4: Predictions on questions about audio relatedness. K: Keyword-based Audio Sensing, S: Semantic Neural Estimator, H: human rating, $T$: true, $F$: false.

## 5.5 Qualitative Results

Figure 9 illustrates the HEAR's responses to audio-related question. HEAR is given the question "what genre of music is playing?" as the audio-related question, and it generates the answer sentence "I can't be sure, but I think it's punk music." Here, HEAR precisely predicted that the given audio was punk music despite the challenging work of discerning what the sound is in the video. The more interesting fact is that HEAR represents its opinion as "I can't be sure" about predicting music. When we listened to that audio in Figure 9 (a), it was really quiet and difficult to identify what the sound is. We guess the HEAR also learned the knowledge about which sounds are difficult for humans to distinguish. Table 5 shows predictions on questions by Keyword-based Audio Sensing and Semantic Neural Estimator. As the keyword-based approach could not understand the meaning of the question, it shows incorrectness in some questions (*i.e.*, (c,d,e)). The neural estimator presents the score $0 < r < 1$ denoting whether the question is related to the audio, which provides proper distinctions between audio-related questions (*i.e.*, (a,b,c,d)) and the others (*i.e.*, (e,f)) based on the meaning of the question.

## 6 Conclusion

We propose Hearing Enhanced Audio Response (HEAR) framework for Video-grounded Dialogue systems. HEAR proposes Sensible Audio Listening and Reconstructive Listening Enhancement, which solve the deaf response problem of VGD systems and validate model-agnostic effectiveness on top of current VGD systems.

## Acknowledgements

This work was supported by Institute for Information & communications Technology Promotion(IITP) grant funded by the Korea government(MSIT) (No. 2021-0-01381, Development of Causal AI through Video Understanding and Reinforcement Learning, and Its Applications to Real Environments) and partly supported by a grant of the KAIST-KT joint research project through AI2XL Laboratory, Institute of convergence Technology, funded by KT [Project No. G01220646, Visual Dialogue System: Developing Visual and Language Capabilities for AI-Based Dialogue Systems].

## Limitations

Our research aims to enhance the audibility of Video-grounded Dialogue (VGD) systems, where we devise Hearing Enhanced Audio Response (HEAR) Framework. As a limitation of our work, we are more focused on the methodology. We need to have a better understanding of the limitation of the proposed method to overcome any failure cases. As shown in the failure case in Appendix, our proposed method has limitations, and we would need to consider other architecture and training methods to incorporate all the necessary information that speech holds. To understand human speech, current audio features seem necessary to be trained more by large-scale audio speech recognition datasets (Panayotov et al., 2015; Garofolo, 1993). Furthermore, although our proposed HEAR mitigates this problem in a methodological idea, we also think that the deaf problem can also be cured by expanding the audio feature dimension (*i.e.*, 128) up to a comparable scale with video (*i.e.*, 4096), such that they include more detailed information. To be specific, we are currently changing the current audio feature extractor (*i.e.*, VGGish) into wav2vec 2.0 (Baevski et al., 2020), which can provide a larger dimensional audio feature (*i.e.*, 768 dimension). We

will also make the audio features (*i.e.*, wav2vec 2.0 features) publicly available and perform a further study on this as our future work.

## Ethics Statement

Video-grounded Dialogue system is one of the conversational AI agents, which is designed to provide assistance to various subsections of our environments including security, entertainment, education, and visual impairments. Our proposed Hearing Enhanced Audio Response framework contributes to improving responses to queries about audio by enhancing the audibility of VGD system. Recently chatbot systems (*e.g.*, ChatGPT) has shown overwhelming performance, as such, we should also think about the potential negative societal impact of these systems. In this respect, we came up with two negative impacts: (1) unreliable vague information by conversational agents and (2) fairness issues in agents' responses. Therefore, word sense disambiguation techniques (Yoon et al., 2022a) and multimodal debiasing solutions (Yoon et al., 2022b; Niu et al., 2021) should also be applied to the dialogue systems.

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

## A  Experimental Details

**Training.**  Proposed HEAR is trained on NVIDIA Quadro RTX 8000 (48GB of memory) GPU. The optimization details are as follows. The AdamW optimizer (Loshchilov and Hutter, 2017) is used with parameters as $\beta_1 = 0.9$, $\beta_2 = 0.999$, and $\epsilon = 10e{-}8$. Due to the efficient usage of resources, the previous 3 question-answer pairs (*i.e.*, $\{C, (Q^{r-3}, A^{r-3}), (Q^{r-2}, A^{r-2}), (Q^{r-1}, A^{r-1})\}$) as a dialogue history are introduced into HEAR for answering current questions $Q^r$. For the learning rate, we first set it as $lr = 6.24e - 5$ and it linearly decreases following a piecewise linear curve up to $lr = 3.63e - 10$ and the model is trained the model during 15 epochs. The hyperparameters are $\delta = 0.05$ for the margin in $\mathcal{L}_{RLE}$. For the joint $d$-dimensional space, all modalities (*i.e.*, video, audio, words) are embedded $d = 768$ dimensional space. The best model is decided by the lowest loss of the validation set on AVSD@DSTC7 (*i.e.*, AVSD@DSTC8 contains only test set for evaluation.). It takes 14 hours to finish the training and the model is fully optimized in about 10 hours.

**Inference.**  In the inference, answer word tokens are generated in a sequence with a probability, where the beam search is applied to avoid prompt word selection with a beam size of 5. The max length for the answer word tokens is set to 20 with a length penalty of 0.3. In the Experiment of the main paper, every performance of HEAR is averaged 15 times with a random seed number.

## B  Additional Experiments

To improve the reproducibility of proposed modules (*i.e.*, SAL, RLE) in HEAR, we give detailed explanations about them with additional results and illustrations of the method.

### B.1  Keyword lists for keyword-based audio sensing

For details of Section 4.3, we list the keywords that we used for keyword-based audio sensing as given below:

$$
\begin{aligned}
\mathrm{W}_{key} = \{ & \text{noise}, \text{sound}, \text{voice}, \text{speech}, \\
& \text{speak}, \text{talk}, \text{listen}, \text{hear}, \text{say}, \text{sing}, \\
& \text{music}, \text{audio}, \text{call}, \text{hum}, \text{loud}, \\
& \text{tones}, \text{utter}, \text{volume}, \text{song}\},
\end{aligned}
\tag{11}
$$

where we also considered all the keywords' plural forms in $\mathrm{W}_{key}$. Figure 10 summarizes the propor-

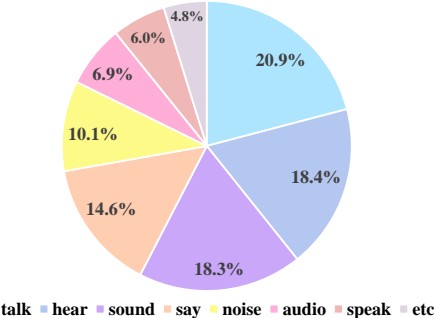

Proportion of audio-related questions according to keywords

Figure 10: Chart for representing the proportion of audio-related questions according to keywords in $\mathrm{W}_{key}$.

| Number of keywords | BLEU 1 | ROUGE-L | CIDEr |
|---|---|---|---|
| 17 | 0.327 | 0.367 | 1.458 |
| 18 | 0.333 | 0.368 | 1.466 |
| 19 | 0.335 | 0.371 | 1.469 |
| 20 | 0.331 | 0.366 | 1.468 |
| 21 | 0.324 | 0.363 | 1.465 |
| 22 | 0.321 | 0.361 | 1.460 |

Table 5: Performance variations according to the number of keywords for Keyword-based Audio Sensing on AVSD@DSTC7 validation set

tion of audio-related questions according to each keyword, where 'talk', 'hear', and 'sound' are the most related words to the audio questions. We additionally considered more keywords (*e.g.*, 'tell', 'background'), but they rather degrade the motivation of keyword-based audio sensing by finding other questions (*e.g.*, 'can you tell me...', 'can you see something in the background') excluding audio-related questions. To be specific, we extend the evaluations for SAL in terms of Keyword-based Audio Sensing (Table A). We investigate the performance variations according to the quantity of audio-related keywords utilized. The results show a positive correlation between the number of keywords and the enhancement. However, surpassing a specific number (*i.e.*, 19) leads to a reversal in this trend, revealing a detrimental correlation. This implies that the pool of keywords starts to include words that lack sensibility in effectively categorizing audio-related questions. Thus, we employ 19 keywords for SAL listed in Appendix B.1

### B.2  Surrounding Masking Scheduling

For the reconstruction upper bound in Section 4.4, we provide further ablation studies about distance scheduling for surrounding masking. The surrounding mask is designed for promoting ranking loss in

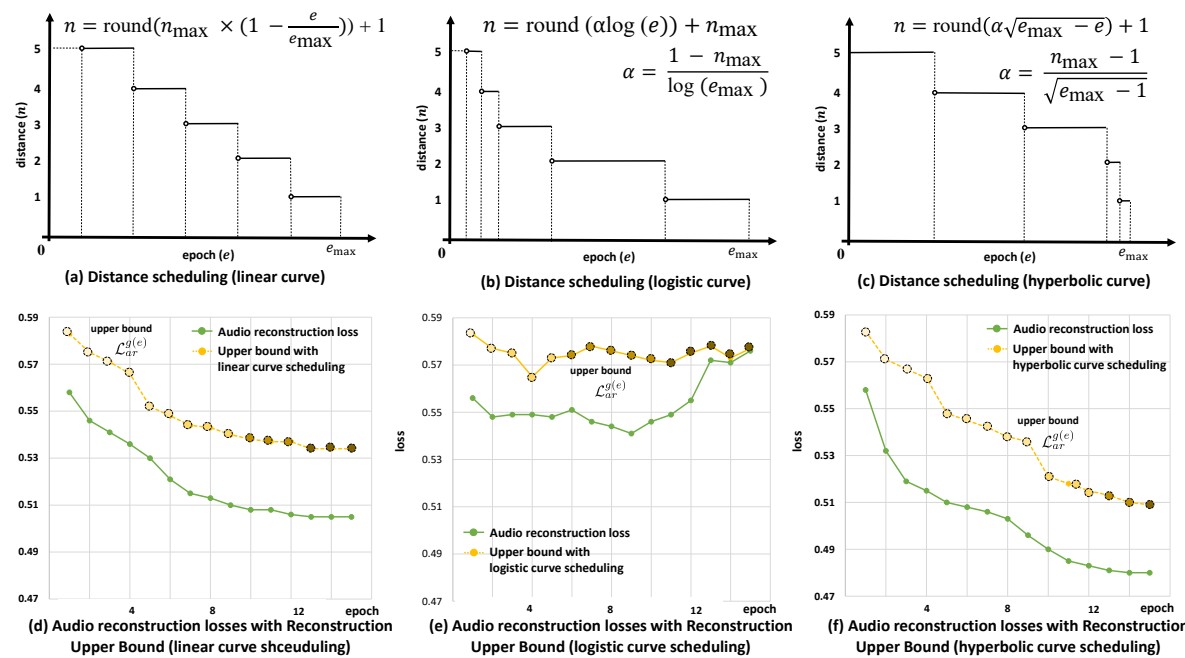

Figure 11: Illustrations of distance scheduling for surrounding masking. The distance $n$ decides how much mask video and audio features for establishing upper bound $\mathcal{L}_{ar}^{n}$: (a) linear curve, (b) logistic curve, (c) hyperbolic curve. Audio reconstruction losses with reconstruction upper bound using (d) linear curve scheduling, (e) logistic curve scheduling, (f) hyperbolic curve scheduling.

$\mathcal{L}_{rub}$ by ensuring that the audio reconstruction $\mathcal{L}_{ar}$ based on the surrounding information has an improved quality than audio reconstruction $\mathcal{L}_{ar}^{n}$ without the surrounding information. To make more effective $\mathcal{L}_{rub}$, $\mathcal{L}_{ar}^{n}$ can be designed to minimize its loss by By reducing the distance $n$ of the surrounding mask. To this end, we leverage the extent of the surrounding mask under our designed various modelings $n = g(e)$ for masking scheduling in Figure 11. We narrowed down the extent of surrounding masking based on three different curves: linear curve, logistic curve, and hyperbolic curve. The hyperbolic curve and linear curve show effectiveness in optimizing validation loss. However, the logistic curve shows some deterioration in the optimization. We think this is because the logistic curve makes the surrounding masking applied to a very narrow area from the beginning of training, which acts like a hard negative that is almost similar to the positive reconstruction (*i.e.*, reconstruction from only masked target audio). Providing hard negatives in early training is considered to hinder optimization when the learnable weights of the model are not properly trained. Therefore, surrounding masking with hyperbolic scheduling is the most effective.

Table 6 summarizes the results of HEAR according to the distance $n$ on the validation split of

| $n_{max}$ | BLEU1 | ROUGE-L | CIDEr |
|---|---|---|---|
| 2 | 0.310 | 0.362 | 1.436 |
| 3 | **0.348** | 0.371 | 1.475 |
| 4 | 0.341 | **0.384** | **1.492** |
| 5 | 0.316 | 0.369 | 1.438 |
| 6 | 0.314 | 0.368 | 1.436 |

Table 6: Ablation study on the distance of negative masking of HEAR on valid split of AVSD@DSTC7.

AVSD@DSTC7 under distance scheduling with the hyperbolic curve. When $n_{max}$ is small (*e.g.*, $n_{max}$ = 2 or 3), $\mathcal{L}_{ar}^{*}$ makes audio reconstruction based on the nearest video and audio from the target audio. This can be effective in terms of improving the connectivity of neighboring video and audio at a narrow distance, however since the distance is too close, there is no significant difference from the positive audio reconstruction, which hinders reconstruction. Based on results in Table 6, negative masking was effective when $n_{max}$ = 4 or 5. In this case, the surrounding modalities' features for audio reconstruction were properly removed by masking, in the meanwhile the surroundings were not masked excessively. Thus the masking contributes on hard negative audio reconstruction. When $n_{max}$ is large (*e.g.*, $n_{max}$ = 6), the perfor-

| Type | HEAR | THAM | RLM | GT |
|------|------|------|-----|-----|
| semantic | 2.41 | 2.15 | 1.81 | 2.86 |
| grammatical | 2.35 | 2.24 | 2.16 | 2.76 |
| fluency | 2.62 | 2.58 | 2.41 | 2.81 |

Table 7: Human evaluation on responses to 50 audio-related questions with respect to semantic adequacy, grammatical correctness, fluency. The scores are assigned as ("1: not at all", "2: neutral", "3: correct"), GT: Ground-Truth response.

mance again degrades. For this case we guess that the negative reconstruction is quite different from the positive, so it would not be an effective negative. These studies were also conducted for a larger $n_{\max}$, but no further improvement was confirmed.

## C Human Evaluation

To assess the improvement of HEAR framework in the responses to audio-related questions, we conduct a Human Evaluation. We selected a total of 50 audio-related questions and generate their responses from 3 models (*i.e.*, HEAR, THAM, RLM). We evaluate all responses based on a scale of 1 to 3, considering three key perspectives: semantic adequacy, grammatical correctness, and fluency. Each score denotes that "1: not at all", "2: neutral", "3: correct". Table 7 shows the human evaluation scores based on 11 evaluators. Evaluators rate the responses based on three categories, where HEAR outperformed recent runner models in all three categories, receiving higher ratings. Furthermore, through empirical validation, we have confirmed the presence of inappropriate answer responses within the Ground-Truth. As a result, the human evaluation scores appropriately reflect this by not consistently obtaining a score of 3.0 or a value close to it.

## D Additional results and Failure case

Figure 12 illustrates additional results of HEAR framework. When presented with the question, "Can you hear the TV?" our HEAR provides an affirmative response, stating, "Yes, I can hear the TV." This response is generated by considering both the audio input of the TV and the visual context of the TV image. Although our proposed HEAR improves the understanding of audio, there were instances of failure when it comes to questions pertaining to speech recognition. As shown in Figure 13, the HEAR framework demonstrates lim-

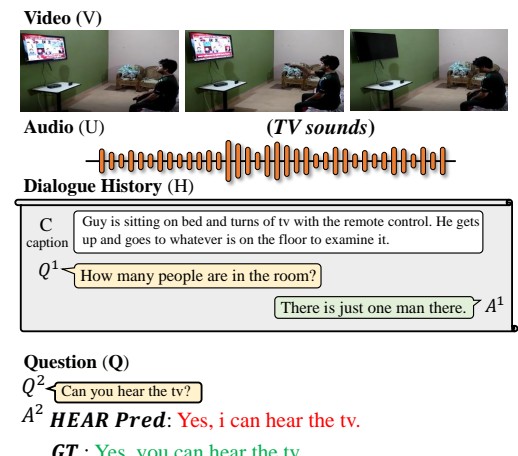

Figure 12: Illustration of additional results of HEAR

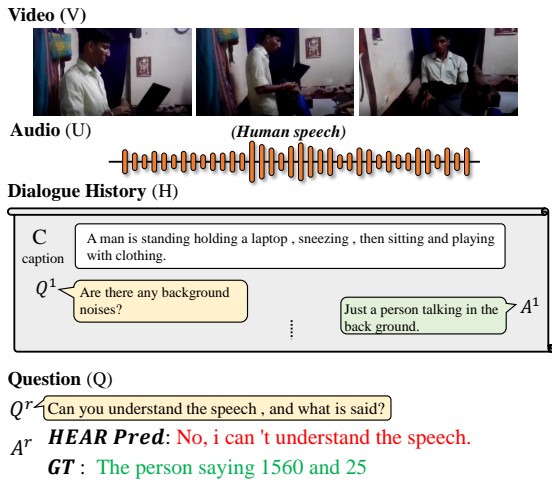

Figure 13: Illustration of failure case of HEAR

itations in accurately comprehending the language within audio inputs. Our empirical studies found that this challenge is prevalent among various dialogue language models, including our model. It is noted that the current audio feature used in the HEAR framework is pre-trained with the environmental sounds dataset, such as AudioSet (Gemmeke et al., 2017). However, it appears that further data training is necessary to enhance the system's ability to understand speech effectively. To address this challenge, our future work will focus on incorporating audio features, such as wav2vec 2.0 (Baevski et al., 2020), which have been trained specifically for audio speech recognition tasks. By leveraging these advanced audio features, we aim to enhance the model's ability to accurately understand and process speech-related information.