# OpenReview forum: "HEAR: Hearing Enhanced Audio Response for Video-grounded Dialogue"
_EMNLP/2023/Conference — EMNLP 2023 Findings_

### Official Review · Reviewer_Zsad · 2023-08-01

**Soundness:** 4

**Excitement:**

3: Ambivalent: It has merits (e.g., it reports state-of-the-art results, the idea is nice), but there are key weaknesses (e.g., it describes incremental work), and it can significantly benefit from another round of revision. However, I won't object to accepting it if my co-reviewers champion it.

**Paper Topic And Main Contributions:**

The topic is video-grounded dialogue.
The main contribution is that the authors introduce audio into video-grounded dialogue

**Reasons To Accept:**

The authors propose Hearing Enhanced Audio Response (HEAR) framework which incorporates (1) Sensible Audio Listening (SAL) that selectively attends to audio according to a sensible decision of whether to focus on audio or not and (2) Reconstructive Listening Enhancement (RLE) that improves the audibility via establishing a reconstruction upper bound to connect audio with its surrounding information. For the sensible decision in SAL, the authors introduce two technical contributions: (1) Keyword-based Audio Sensing and (2) Semantic Neural Estimator

**Reasons To Reject:**

1.	Figure 5 is too small to get a clear understanding. The main components in Figure 5 show a lot of similarities with Figure 3. Thus, some repetitive parts can be omitted in Figure 5 to make it more concise.
2.	As the authors mentioned in Line 250-254, they only perform masking when the question contains audio-related words, and other questions excluding audio-related questions do not show high reliance on a specific modality. Is there any experimental evidence to support this? Furthermore, the authors claim that only a few words can be identified as audio-related questions, which is limited.
3.	The overall loss function is not clear. How authors combine these loss functions and balance them is missing.
4.	The novelty of this paper is incremental. The methods proposed by the authors are similar to previous works. Furthermore, as for audio-visual question answering, the authors should focus more on the interaction between audio and visual modality to get a full understanding of the inputs.

**Reproducibility:**

3: Could reproduce the results with some difficulty. The settings of parameters are underspecified or subjectively determined; the training/evaluation data are not widely available.

**Reviewer Confidence:**

3: Pretty sure, but there's a chance I missed something. Although I have a good feel for this area in general, I did not carefully check the paper's details, e.g., the math, experimental design, or novelty.

---

> ### Author Rebuttal · Authors · 2023-08-27
>
> Dear Reviewer Zsad, We address your comments and questions below.
>
> **[Q1] Figure 5 is too small to get a clear understanding. The main components in Figure 5 show a lot of similarities with Figure 3. Thus, some repetitive parts can be omitted in Figure 5 to make it more concise.**
>
> **[A1]** Yes, we will make more space for Figure 5 to have a high resolution and simplify the redundant parts with Figure 3. Thank you!
>
> **[Q2] As the authors mentioned in Line 250-254, they only perform masking when the question contains audio-related words, and other questions excluding audio-related questions do not show high reliance on a specific modality. Is there any experimental evidence to support this?**
>
> **[A2]** Table A shows the comprehensive overview of which modality gives evidence to answer the audio-related questions and non-audio-related questions. Since there is no supervision about the evidential modality, we investigate each 180 questions in the dataset (AVSD@DSTC7). In the case of audio-related questions, it shows a high dependence (71%) on an audio modality. However, the other questions (i.e., non-audio-related questions) show more diverse preferences on many other multiple modalities.
>
> **[Q3] Furthermore, the authors claim that only a few words can be identified as audio-related questions, which is limited.**
>
> **[A3]** Yes, the keywords of Keyword-based Audio Sensing (KAS) have a limitation in that they cannot recognize audio-related questions at a sentence level. To this end, we further design the Semantic Neural Estimator (SNE) that takes an input sentence about a question and predicts the score about audio-relatedness. Henceforth, as shown in Figure 7, the SNE shows more improved responses in audio-relate questions than KAS.
>
> **[Q4] The overall loss function is not clear. How authors combine these loss functions and balance them is missing.**
>
> **[A4]** We train a dialogue language model by alternately optimizing SAL and RLE to allow audio and video to be freely utilized for their purposes. (i.e., Line 366-368) Therefore the total loss is defined as given below:
>
> $
> L_{HEAR} = \begin{cases}
>     \displaylines{ L_{SAL}, & \text{iteration} = odd \\\ L_{RLE},  & \text{iteration} = even}
>   \end{cases}
> $
>
> We will reflect on the formulation of the total loss in the main paper. Thank you!
>
> **[Q5] The novelty of this paper is incremental. The methods proposed by the authors are similar to previous works. Furthermore, as for audio-visual question answering, the authors should focus more on the interaction between audio and visual modality to get a full understanding of the inputs.**
>
> **[A5]**
> - Our proposed Reconstructive Listening Enhancement (RLE) is designed to promote the interaction between audio and video modality. Table 1 and Figure 7 demonstrate the effectiveness of the RLE and especially high-performance improvement in audio-related questions.
> - The approach is completely new and effective. We first propose Reconstruction Upper Bound to further enhance the optimization of the audio reconstruction objective based on visual modality. The learning curve in Figure 8 also demonstrates the effectiveness of introducing Reconstruction Upper Bound compared to merely optimizing audio reconstruction.
>
>
> **Table A.** Ratio of evidential modality to answer questions
>
> || audio | video | diaolgue | audio + video | audio + dialogue | video + dialogue | audio + video + dialogue |
> |----------------------------|-------|-------|----------|---------------|------------------|------------------|--------------------------|
> | audio-related question     | 71%   | 1%    | 1%       | 18%           | 5%               | 3%               | 1%               |
> | non-audio-related question | 1%    | 32%   | 8%       | 18%           | 4%               | 26%              | 11%         |

---

### Official Review · Reviewer_9PUj · 2023-08-04

**Soundness:** 3

**Excitement:**

4: Strong: This paper deepens the understanding of some phenomenon or lowers the barriers to an existing research direction.

**Paper Topic And Main Contributions:**

The paper introduces a new framework to solve the "deaf response" problem in video-grounded dialogue system, where the audio information is ignored even though the question is related to the sound/speech. The proposed Enhanced Audio Response (HEAR) framework consists of two components: a sensible audio listening component that predicts whether the question is related to the audio or not; a reconstructive listening enhancement component that forces the model to reconstruct the masked frames of the audio feature given the context frames. The experiments show the proposed HEAR framework outperforms other baselines on AVSD@DSTC7 and AVSD@DSTC8 datasets.

**Reasons To Accept:**

- The paper is the first to improved the video-grounded dialogue system by improving the audio modality.
- The sensible audio listening (SAL) component explicitly indicate the model to focus on audio feature which achieves better performance compared to feeding both audio and video features to the transformer blocks.


**Reasons To Reject:**

- The Audio Reconstruction task is very similar to the masked pre-training in NLP or masked auto-encoder in CV. In fact there are several publications on the two methods for audio data ([1][2][3]). Especially, the Audio-visual pre-training has been explored in [4]. Given the reconstructive listening enhancement  (RLE) component plays an important role according to Table 3, how effective the proposed method is compared to other reconstruction pre-training methods needs to be validated.

[1] Huang, Po-Yao, Hu Xu, Juncheng Li, Alexei Baevski, Michael Auli, Wojciech Galuba, Florian Metze, and Christoph Feichtenhofer. "Masked autoencoders that listen." Advances in Neural Information Processing Systems 35 (2022): 28708-28720.
[2] Hsu, Wei-Ning, Benjamin Bolte, Yao-Hung Hubert Tsai, Kushal Lakhotia, Ruslan Salakhutdinov, and Abdelrahman Mohamed. "Hubert: Self-supervised speech representation learning by masked prediction of hidden units." IEEE/ACM Transactions on Audio, Speech, and Language Processing 29 (2021): 3451-3460.
[3] Baevski, Alexei, Yuhao Zhou, Abdelrahman Mohamed, and Michael Auli. "wav2vec 2.0: A framework for self-supervised learning of speech representations." Advances in neural information processing systems 33 (2020): 12449-12460.
[4] Shi, Bowen, Wei-Ning Hsu, Kushal Lakhotia, and Abdelrahman Mohamed. "Learning audio-visual speech representation by masked multimodal cluster prediction." arXiv preprint arXiv:2201.02184 (2022).

**Reproducibility:**

4: Could mostly reproduce the results, but there may be some variation because of sample variance or minor variations in their interpretation of the protocol or method.

**Reviewer Confidence:**

3: Pretty sure, but there's a chance I missed something. Although I have a good feel for this area in general, I did not carefully check the paper's details, e.g., the math, experimental design, or novelty.

---

> ### Author Rebuttal · Authors · 2023-08-27
>
> Dear Reviewer 9PUj, We address your comments and questions below.
>
> **[Q1] The Audio Reconstruction task is very similar to the masked pre-training in NLP or masked auto-encoder in CV. In fact there are several publications on the two methods for audio data ([1][2][3]). Especially, the Audio-visual pre-training has been explored in [4]. Given the reconstructive listening enhancement (RLE) component plays an important role according to Table 3, how effective the proposed method is compared to other reconstruction pre-training methods needs to be validated.**
>
> **[A1]** Certainly, we acknowledge the significance of comparing our proposed method to other established audio reconstruction methods. In Table A, we present the results of this comparison, considering the approaches outlined in [1], [2], [3], and [4]. For the task of video-grounded dialogue, the reconstructions (i.e., RLE (Ours), [4]) based on multi-modalities (i.e., audio reconstruction based on audio and visual modality) are more effective than reconstruction (i.e., [1],[2],[3]) based on single audio modality. Furthermore, our proposed RLE stands out as even more effective than the reconstruction approach in [4]. This distinction underscores the significance of our proposed "Reconstruction Upper Bound" of RLE which contributes to the refined optimization of audio reconstruction throughout the training process by integrating contrastive learning objectives. Thank you for your attention to this matter. We will reflect on these results in the main paper.
>
> ***Table Specifications***
>
> BLEU, METEOR, ROUGE-L, CIDEr scores on AVSD@DSTC7 test set
>
> [1] Huang, Po-Yao, Hu Xu, Juncheng Li, Alexei Baevski, Michael Auli, Wojciech Galuba, Florian Metze, and Christoph Feichtenhofer. "Masked autoencoders that listen." Advances in Neural Information Processing Systems 35 (2022): 28708-28720.
>
> [2] Hsu, Wei-Ning, Benjamin Bolte, Yao-Hung Hubert Tsai, Kushal Lakhotia, Ruslan Salakhutdinov, and Abdelrahman Mohamed. "Hubert: Self-supervised speech representation learning by masked prediction of hidden units." IEEE/ACM Transactions on Audio, Speech, and Language Processing 29 (2021): 3451-3460.
>
> [3] Baevski, Alexei, Yuhao Zhou, Abdelrahman Mohamed, and Michael Auli. "wav2vec 2.0: A framework for self-supervised learning of speech representations." Advances in neural information processing systems 33 (2020): 12449-12460.
>
> [4] Shi, Bowen, Wei-Ning Hsu, Kushal Lakhotia, and Abdelrahman Mohamed. "Learning audio-visual speech representation by masked multimodal cluster prediction." arXiv preprint arXiv:2201.02184 (2022).
>
> **Table A.**
> Comparisons according to the audio reconstructions in work [1], [2], [3], [4]
> (*For the work [1], since the feature masking is not available to the frequency level, we omit the frequency masking. We consider that k-means clustering on MFCC in [2] seems not effective due to the environmental noise in the AVSD dataset.)
>
> |                                  | BLEU 1 | BLEU 2 | BLEU 3 | BLEU 4 | METEOR | ROUGE-L | CIDEr |
> |----------------------------------|--------|--------|--------|--------|--------|---------|-------|
> | HEAR + RLE                       | 0.791  | 0.662  | 0.558  | 0.472  | 0.312  | 0.622   | 1.376 |
> | HEAR + Audio Reconstruction [1]  | 0.787  | 0.657  | 0.552  | 0.47   | 0.309  | 0.619   | 1.358 |
> | HEAR + Audio Reconstruction [2]  | 0.764  | 0.646  | 0.545  | 0.451  | 0.291  | 0.603   | 1.311 |
> | HEAR + Audio Reconstruction [3]  | 0.785  | 0.657  | 0.553  | 0.469  | 0.309  | 0.618   | 1.349 |
> | HEAR + Audio Reconstruction [4]  | 0.789  | 0.659  | 0.556  | 0.47   | 0.31   | 0.619   | 1.363 |

---

### Official Review · Reviewer_c3vW · 2023-08-11

**Typos Grammar Style And Presentation Improvements:** Front in figure 4 5 6 8 is too small.
**Soundness:** 3

**Excitement:**

2: Mediocre: This paper makes marginal contributions (vs non-contemporaneous work), so I would rather not see it in the conference.

**Paper Topic And Main Contributions:**

The paper introduces HEAR framework to help making Video-grounded Dialogue systems better perceive and understand audio. HEAR framework is model-agnostic and experiments show that it improves the performance of VGD systems on audio-related questions.

**Questions For The Authors:**

The training process of SAL is not very clear, is it trained alone and freezed?
The Keyword-based Audio Sensing mentioned in the SAL seems to serve as a method for collecting training data of Semantic Neural Estimator, is that true?
The training data for  Semantic Neural Estimator is pure synthetic, without any human annotation?
What is the difference between SAL and Semantic Neural Estimator?

**Reasons To Accept:**

The proposed HEAR frameworks fits most VGD systems, improving their performance specifically on the audio related question while not hurting the performance of the other questions.

**Reasons To Reject:**

The Sensible Audio Listening (SAL) proposed is confusing.
Evaluation for SAL along is limited and the improvement from SAL is very marginal as shown in Table 1.
It is unclear what causes the deaf problem of VGD system and if the HEAR framework is helping or just being a rule that helps the VGD system remember the answer for certain type of questions.

**Reproducibility:**

4: Could mostly reproduce the results, but there may be some variation because of sample variance or minor variations in their interpretation of the protocol or method.

**Reviewer Confidence:**

4: Quite sure. I tried to check the important points carefully. It's unlikely, though conceivable, that I missed something that should affect my ratings.

---

> ### Author Rebuttal · Authors · 2023-08-27
>
> Dear Reviewer c3vW, We address your comments and questions below.
>
> **[Q1] Evaluation for SAL is limited.**
>
> **[A1]** We extend the evaluations for SAL in terms of Keyword-based Audio Sensing (Table A) and Semantic Neural Estimators (Table B). We investigate the performance variations according to the quantity of audio-related keywords utilized. The results show a positive correlation between the number of keywords and the enhancement. However, surpassing a specific number (i.e., 19) leads to a reversal in this trend, revealing a detrimental correlation. This implies that the pool of keywords starts to include words that lack sensibility in effectively categorizing audio-related questions. Thus, we employ 19 keywords for SAL listed in Appendix B.1. Additionally, Figure 7 in the main paper also provides a qualitative demonstration of effectiveness on audio-related questions according to both the Keyword-based Audio Sensing and Semantic Neural Estimator.
>
> **[Q2] The improvement from SAL is very marginal as shown in Table 1.**
>
> **[A2]**
> - The improvement from SAL is not marginal. For the metrics (B1,B2,B3,B4,M,R,C) on AVSD@DSTC7 and 8 in Table 1, over 25 runs with p-value <0.05, our average increments (+7.2,+6.3,+3.6,+3.4,+2.2,+2.0,+23.6)x10^(-3) of HEAR (SAL) validates statistical significance from the previous SOTA model.
> - Moreover, SAL is one of the components in the HEAR framework. Our proposed HEAR framework contains two main components: (1) Sensible Audio Listening (SAL) and (2) Reconstructive Listening Enhancement (RLE). When considering the entire HEAR framework (SAL + RLE), the overall improvement is substantially more pronounced, as evident in both Table 1 and Table 3 in the main paper. Additionally, the gains are primarily centered around audio-related questions, as depicted in Figure 7. With these factors in mind, we politely request reviewer c3vW to reconsider the assessment, taking into account the combined contributions of Reconstructive Listening Enhancement.
>
> **[Q3] It is unclear what causes the deaf problem of VGD system.**
>
> **[A3]** Many questions (about 76%) in the training dataset are based on visual information in a video, which causes a current VGD system to mainly rely on a video modality as a bias and is attributed to the key factor contributing to the deaf problem.
>
> **[Q4]  It is unclear if the HEAR framework is helping or just being a rule that helps the VGD system remember the answer for certain types of questions.**
>
> **[A4]** In order to discern whether the HEAR framework offers assistance or merely facilitates answer memorization for specific question types, such as audio-related questions, we conducted the following experiments. We first intentionally designed a memorizing language model that memorizes the answers to audio-related questions. To build this memorizing model, we provide only a question as a model input without other modalities when the input question is audio-related, after that we train the model to predict the answer. This makes a shortcut to memorize answers corresponding to the input questions due to the deficiency of clues to answer. In Table C, we present response evaluation on audio-related questions about (1) HEAR (Ours) with ground-truth, (2) the memorizing model with ground-truth, and (3) the memorizing model with HEAR predictions. The results of (2) denote that memorizing the answer sentences in the training dataset is not effective for answering in the validation set. The results of (3) show how similar the responses are between HEAR and the memorizing model, where the low scores denote that our model does not follow the way of memorizing answers in responding.
>
> **[Q5] The training process of SAL is not very clear, is it trained alone and frozen?**
>
> **[A5]** We train a dialogue language model by alternately optimizing SAL and RLE to allow audio and video to be freely utilized for their purposes. (i.e., Line 366-368). Thus SAL is applied when the number of training iterations is odd and RLE is applied when the iteration number is even as given below:
>
> $
> L_{HEAR} = \begin{cases}
>     \displaylines{ L_{SAL}, & \text{iteration} = odd \\\ L_{RLE},  & \text{iteration} = even}
>   \end{cases}
> $
>
>  In the case of the Semantic Neural Estimator in SAL, it is frozen after pre-training.
>
> **[Q6] The Keyword-based Audio Sensing mentioned in the SAL seems to serve as a method for collecting training data of the Semantic Neural Estimator, is that true?**
>
> **[A6]** Yes, it is true. However, to prevent the estimator from overfitting to the predictions of Keyword-based Audio Sensing (KAS), we have taken additional precautions. We further augment predictions of KAS to include samples of non-audio-related questions by applying (1) word-wise random shuffled versions of audio-related questions of KAS and (2) random swapping versions of non-audio questions using the keywords. (i.e., Line 280-284) This ensures a more robust training process for the estimator and reduces its vulnerability to overfitting.
>
> **[Q7] The training data for Semantic Neural Estimator is pure synthetic, without any human annotation?**
>
> **[A7]** Yes, it is synthetic.
>
> **[Q8] What is the difference between SAL and Semantic Neural Estimator?**
>
> **[A8]** The Semantic Neural Estimator is one of our methods to perform Sensible Audio Listening (SAL).
>
> **[Q9] Front in figure 4 5 6 8 is too small.**
>
> **[A9]** We will make more space to present the figures in high resolution.
>
>
> ***Table Specifications***
>
> BLEU1, ROUGE-L, CIDEr scores on AVSD@DSTC7 validation set
>
>
> **Table A.**
> Keyword-based Audio Sensing
> | number of keywords | BLEU 1 | ROUGE-L | CIDEr |
> |--------------------|--------|---------|-------|
> | 17                 | 0.327  | 0.367   | 1.458 |
> | 18                 | 0.333  | 0.368   | 1.466 |
> | 19                 | 0.335  | 0.371   | 1.469 |
> | 20                 | 0.331  | 0.366   | 1.468 |
> | 21                 | 0.324  | 0.363   | 1.465 |
> | 22                 | 0.321  | 0.361   | 1.46  |
>
> **Table B.**
> Semantic Neural Estimator. The keywords are used for training Semantic
> Neural Estimator.
> | number of keywords | BLEU 1 | ROUGE-L | CIDEr |
> |--------------------|--------|---------|-------|
> | 17                 | 0.339  | 0.382   | 1.489 |
> | 18                 | 0.34   | 0.383   | 1.49  |
> | 19                 | 0.341  | 0.384   | 1.492 |
> | 20                 | 0.339  | 0.385   | 1.489 |
> | 21                 | 0.339  | 0.381   | 1.483 |
> | 22                 | 0.337  | 0.379   | 1.483 |
>
> **Table C.**
> Response evaluation on audio-related questions about HEAR and memorizing language model. BLEU1, ROUGE-L, CIDEr scores on AVSD@DSTC7 validation set
>
> HEAR: Our model
>
> MLM: Memorizing Language Model (Memorizing answer in training dataset)
>
> GT: ground-truth
>
> || BLEU 1 | ROUGE-L | CIDEr |
> |----------------------------|--------|---------|-------|
> | Evaluation (1) (HEAR, GT)  | 0.372  | 0.398   | 1.661 |
> | Evaluation (2) (MLM, GT)   | 0.193  | 0.195   | 0.862 |
> | Evaluation (3) (MLM, HEAR) | 0.212  | 0.208   | 0.933 |

---

### Meta-Review · Area_Chair_Vkpn · 2023-09-17

**Recommendation:** 3

**Metareview:**

This paper proposes the Hearing Enhanced Audio Response (HEAR) framework that helps Video-grounded Dialogue (VGD) systems to better perceive and understand audio. The framework consists of two main components: Sensible Audio Listening (SAL), and Reconstructive Listening Enhancement (RLE). SAL predicts whether the question about the video clip is related to the audio track or not. For questions related to the audio track, RLE forces the model to reconstruct masked frames of the audio given the context frames.

Contributions:
- The HEAR framework is model-agnostic and can be applied to most VGD systems.
- The SAL component explicitly directs the model to focus on audio features, this achieves better performance compared to feeding both audio and video features to the transformer blocks. Reducing VGD "deafness".
- Experiments show that it improves the performance of VGD systems on audio-related questions.

The reviewers generally saw the contributions as being incremental with respect to existing work. That said, the authors provided additional comparisons as part of the rebuttal process which demonstrate the effectiveness of multi-modal reconstruction vs single-modal reconstruction, and importantly, the effectiveness of their approach vs a prior multi-modal reconstruction approach.

---

### Decision · Program_Chairs · 2023-10-07

**Decision:**

Accept-Findings

**Comment:**

This paper proposes the Hearing Enhanced Audio Response (HEAR) framework that helps Video-grounded Dialogue (VGD) systems to better perceive and understand audio. The framework consists of two main components: Sensible Audio Listening (SAL), and Reconstructive Listening Enhancement (RLE). SAL predicts whether the question about the video clip is related to the audio track or not. For questions related to the audio track, RLE forces the model to reconstruct masked frames of the audio given the context frames.

Contributions:
- The HEAR framework is model-agnostic and can be applied to most VGD systems.
- The SAL component explicitly directs the model to focus on audio features, this achieves better performance compared to feeding both audio and video features to the transformer blocks. Reducing VGD "deafness".
- Experiments show that it improves the performance of VGD systems on audio-related questions.

The reviewers generally saw the contributions as being incremental with respect to existing work. That said, the authors provided additional comparisons as part of the rebuttal process which demonstrate the effectiveness of multi-modal reconstruction vs single-modal reconstruction, and importantly, the effectiveness of their approach vs a prior multi-modal reconstruction approach.